# Stingray Venom Proteins: Mechanisms of Action Revealed Using a Novel Network Pharmacology Approach

**DOI:** 10.3390/md20010027

**Published:** 2021-12-24

**Authors:** Kim N. Kirchhoff, André Billion, Christian R. Voolstra, Stephan Kremb, Thomas Wilke, Andreas Vilcinskas

**Affiliations:** 1Department of Animal Ecology and Systematics, Justus Liebig University Giessen, Heinrich-Buff-Ring 26-32, 35392 Giessen, Germany; Tom.Wilke@allzool.bio.uni-giessen.de; 2Department of Bioresources, Fraunhofer Institute for Molecular Biology and Applied Ecology, Ohlebergsweg 12, 35392 Giessen, Germany; andre.billion@ime.fraunhofer.de; 3Department of Biology, University of Konstanz, 78457 Konstanz, Germany; christian.voolstra@uni-konstanz.de; 4Red Sea Research Center, Division of Biological and Environmental Science and Engineering (BESE), King Abdullah University of Science and Technology (KAUST), Thuwal 23955-6900, Saudi Arabia; stephan.kremb@gmail.com; 5LOEWE Centre for Translational Biodiversity Genomics (LOEWE-TBG), Senckenberganlage 25, 30325 Frankfurt am Main, Germany; 6Institute for Insect Biotechnology, Justus Liebig University of Giessen, Heinrich-Buff-Ring 26-32, 35392 Giessen, Germany

**Keywords:** transcriptomics, high-content screening, multi-omics data integration, drug discovery, venomics

## Abstract

Animal venoms offer a valuable source of potent new drug leads, but their mechanisms of action are largely unknown. We therefore developed a novel network pharmacology approach based on multi-omics functional data integration to predict how stingray venom disrupts the physiological systems of target animals. We integrated 10 million transcripts from five stingray venom transcriptomes and 848,640 records from three high-content venom bioactivity datasets into a large functional data network. The network featured 216 signaling pathways, 29 of which were shared and targeted by 70 transcripts and 70 bioactivity hits. The network revealed clusters for single envenomation outcomes, such as pain, cardiotoxicity and hemorrhage. We carried out a detailed analysis of the pain cluster representing a primary envenomation symptom, revealing bibrotoxin and cholecystotoxin-like transcripts encoding pain-inducing candidate proteins in stingray venom. The cluster also suggested that such pain-inducing toxins primarily activate the inositol-3-phosphate receptor cascade, inducing intracellular calcium release. We also found strong evidence for synergistic activity among these candidates, with nerve growth factors cooperating with the most abundant translationally-controlled tumor proteins to activate pain signaling pathways. Our network pharmacology approach, here applied to stingray venom, can be used as a template for drug discovery in neglected venomous species.

## 1. Introduction

More than 200,000 animal species produce venom, mostly for defense and/or predation [1,2,3,4]. Venoms are cocktails of up to 3000 bioactive compounds, including protein/peptide toxins, metabolites and salts [3,5]. Venoms are produced in specialized tissues or glands, and are actively transferred to target organisms via spines, teeth or modified cell-harpoons [3]. Venom targets are found in most major physiological pathways, inducing local effects such as tissue disruption and systemic effects such as paralysis [3,6,7,8]. The diverse components of venoms offer a rich source of novel molecular entities for medical applications. The first example was the cardiovascular agent captopril, which was approved in 1981 for the treatment of hypertension and congestive heart failure [9]. Twelve toxin-based drugs are currently approved for indications such as cardiovascular disease, chronic pain and diabetes, and are derived from species with abundant and/or pure venoms, such as snakes, cone snails, and the gila monster [9,10]. Other toxins are used as research tools to identify new pharmaceutical targets and mechanisms of action (MoA), or for the development of pesticides and cosmetics [11,12]. Animal venoms could help to fill the drug development pipeline because there are hundreds of venomous species each producing up to thousands of bioactive molecules, potentially yielding more than 20 million drug candidates [13].

Animal toxins interact with a vast array of targets [8], but MoAs, toxin synergisms and polypharmacology (the ability to act on multiple targets) have been determined for only a few toxins. The MoA is an elementary property for drug discovery and development [14], which can be used to reduce the side effects often responsible for high attrition rates during clinical trials [15], and to facilitate the development of antivenoms [16]. This knowledge gap must be addressed urgently, given the increasing realization that most drugs act on multiple targets [17,18] and participate in synergistic interactions to modulate complex biological systems [18,19].

Network pharmacology is a drug discovery tool that integrates multiple omics methods on a functional level to achieve a holistic interpretation of biological responses to specific molecules [20]. Here we report the development of a novel network pharmacology approach for the comprehensive prediction of animal venom protein MoAs (Figure 1). Most venomous species have been neglected as a source of drug leads because their venoms are inaccessible in sufficient quantity and/or purity for analysis, often being extracted as crude venoms contaminated with mucus and the contents of epidermal cells [21]. In particular, many marine, vertebrate, defensive and non-glandular (secretory cells) venoms have been overlooked [21,22,23]. Modern sample-friendly omics approaches can help to overcome these limitations, making venoms more accessible for drug discovery. More than 30% of venomous vertebrates are fish [1], with defensive venom systems (and diverse delivery structures) arising by convergent evolution at least 19 times among 2900 species [24,25]. However, the remote habitats of these species and the typical impurity of venom samples limit our ability to test venom composition and activity. We selected stingrays as a representative system because they best reflect limitations set out above. Stingrays are the most speciose order of venomous cartilaginous fish, including 218 extant marine, brackish and freshwater species featuring defensive tail spines covered with venom-secreting cells, epidermis and mucus [26,27,28]. To better understand the composition and MoAs of stingray venom proteins, we created a functional data network based on integrated gene expression and bioactivity profiles. We analyzed the transcriptomes of venomous spine tissue isolated from five marine and freshwater stingray species to determine which genes were strongly expressed, and thus most likely to encode venom peptides and proteins. We then determined the effect of stingray venom on HeLa cells by image-based high-content bioactivity screening. In the final network pharmacology step, the transcriptomic and bioactivity data were screened against a functional database to create a large data network for comparative analysis, thus increasing the information content. This network combined the original compositional and bioactivity properties with additional target, pathway and druggability data. Specific, curated clusters were extracted from this network to reveal MoAs associated with envenomation symptoms, target systems and pathways. This network pharmacology approach is an inexpensive sample-friendly alternative to traditional methods, providing a comprehensive insight into MoAs and toxin synergisms of crude and neglected animal venoms.

## 2. Results

### 2.1. Crude Venom Transcripts

RNA was isolated from five stingray species representing the two most important stingray habitats and the two largest families: Dasyatidae (marine, *Dasyatis pastinaca*, *Himantura leoparda*, and *Pteroplatytrygon violacea*) and Potamotrygonidae (fresh water, *Potamotrygon leopoldi*, and *Potamotrygon motoro*). An average of 30.5 million raw reads were assembled into an average of 2.1 million transcripts per species transcriptome. The bioinformatics workflow is shown in Appendix B (Figure A1), the complete datasets are provided in Appendix A, quality parameters are listed in Table A1, and the evaluation is summarized in Appendix A.

Transcriptome-inferred crude venom composition and expression patterns should be considered only as a guideline because replicates are missing, and it is difficult to distinguish between toxins and physiological proteins. Even so, venom composition based exclusively on toxin-related annotations against the ToxProt database indicated high interspecific similarity, with 24% of toxin families overlapping in all five stingray transcriptomes, representing 65–74% of all hits (Figure 2A and Appendix A). This is translated to 19 shared toxin families, among which the translationally-controlled tumor protein (TCTP) family was the most strongly expressed in each species, often directly followed by the glycosyl hydrolase 56 (hyaluronidase) family, although hyaluronidase expression was low in *H. leoparda* and *P. violacea*. The shared family dataset also included the ohanin/vespryn and phospholipase A2 (PLA2) families, and the widely distributed venom kunitz-type, venom metalloproteinase (M12B), and peptidase S1 toxin families (Appendix A). Members of the calmodulin family and several putative neurotoxins representing the snake three-finger toxin (TFT) family were also strongly expressed in stingray venom tissue. However, some toxin-like transcripts matched proteins/peptides with no family attribution such as the augerpeptide hhe53, which was found in all five transcriptomes and was the most abundant transcript in *H. leoparda*, *P. leopoldi* and *P. violacea*. From a biochemical perspective, the stingray venom mainly featured enzymes and other proteins, but has just a low content of peptides (<50 amino acids), representing only ~6% of the annotated hits against ToxProt when fragments were ignored (Appendix A).

We also conducted an interfamilial comparison of the crude venom transcripts between the marine (Dasyatidae) and freshwater (Potamotrygonidae) species (Figure 2B). Differences were only considered when more than one species shared the same pattern. The most remarkable difference was the stronger expression of the true venom lectin family in marine species, contrasting with the low or absent expression in freshwater species.

### 2.2. Venom Bioactivity

Crude venom from three stingray species (*D. pastinaca*, *P. violacea*, and *P. leopoldi*) was tested in a high-content screening (HCS) assay based on HeLa cells. The species were selected based on their comparability in the transcriptome dataset and the availability of sufficient fresh venom samples for bioactivity screenings. The raw data were reduced in a stepwise manner to 31 highly informative cytological parameters (Appendix C). The raw and reduced datasets, as well as lists of reference drugs (clustering with our crude venom extracts), are listed in the Appendix A.

The cytological profiles of all three stingray venom extracts revealed dose-dependent effects against the NF-κB pathway and the cytoskeleton, as well as the nucleus, mitochondria, endoplasmic reticulum (ER) and lysosomes (Figure 2C,D). These results indicated a slight increase in the translocation of NF-κB into the nucleus, the degradation of F-actin, nuclear chromatin condensation, impairment of mitochondrial thiol-related pathways, a loss of ER integrity, and the depletion of lysosomes. In all cases the effects were strongest for the undiluted venom extract of *P. violacea*. The cytological profiles of stingray venom clustered with those of ~29% of all the reference compounds we tested. This revealed that drugs with similar bioactivity profiles to crude stingray venoms (and probably similar MoA) mainly target the nervous and cardiovascular systems, but also the immune system (e.g., anti-inflammatory and anti-allergic drugs).

The comparison of marine and freshwater species revealed that marine but not freshwater stingray venom induces effects against the ER and lysosomes. In addition, *P. violacea* venom triggered the translocation of p53 into the nucleus, membrane disruption (resulting in the cytoplasmic redistribution of the membrane marker), ~40% cell death, cell cycle arrest during the G0/G1 phase, and a strong increase in the number of cells with a low DNA content, indicating damaged (necrotic) or apoptotic cells. Interestingly, 11 of 12 reference compounds clustering with the (marine) *D. pastinaca* venom profile overlapped with those clustering with (marine) *P. violacea* venom but none of them overlapped with (freshwater) *P. leopoldi* venom.

### 2.3. Network Pharmacology

The transcriptome and bioactivity datasets each offer a limited perspective on the MoA of stingray toxins. However, the integration of both data topologies at the same functional level allows a more holistic interpretation. Therefore, stingray toxin-like transcripts and reference compounds with similar MoAs to our stingray venom extracts were screened against the KEGG database, linking the transcripts and clustered drugs to molecules with known targets, target systems, pathways and associated diseases. An average of 32% of all stingray transcripts and 58% of clustered reference compounds were annotated in this manner. The datasets were integrated at a pathway level because this was the most informative and abundantly inferred functional category. The pathways inferred by transcript and reference compound annotation were compared, resolving the integrated dataset to a total of 216 pathways among which 29 were shared and represented 15 target systems, including the *Nervous system*, *Circulatory system*, *Cancer*, *Signal Transduction*, and the *Immune system*. We focused on this shared dataset, which included 70 transcripts representing 16 toxin families and 70 reference drugs clustered by HCS mostly neuropsychiatric and cardiovascular agents. The core dataset included the interlaced toxin-like transcripts, cellular bioactivities, and functional data enabling the prediction of stingray toxins MoAs against individual target systems and pathways in the context of specific envenomation outcomes.

Fish venom triggers severe pain as the primary envenomation symptom, and we congruently identified nine pathways in the core dataset with potential roles in pain. These pathways represented the *Signal transduction*, *Nervous system* and *Sensory system* categories and were targeted by 20 transcripts and 41 HCS-clustered reference drugs (Appendix D). The manual curation and graphical representation of molecular targets and associated signaling cascades in the pain-related pathways allowed us to predict the MoA of pain-inducing stingray venom components in the nervous system, which we refer to hereafter as the pain cluster (Figure 3).

Our pain cluster clearly indicated that G-protein coupled receptor (GPCR) calcium signaling is a key MoA for pain-inducing stingray toxins. Specifically, the transcriptome hits bibrotoxin and cholecystoxin, and 16 of the HCS hits (including ebastine, ekomine, and carvedilol) were found to target distinct GPCRs (5-HTR2, HRH1, ENDR, CC2KR, or ADR), subsequently activating a Gα protein (Gαq). This in turn causes phospholipase C (PLC) to hydrolyze a membrane substrate into inositol-1,4,5-triphosphate (IP3) and diacylglycerol (DAG). IP3 acts as a second messenger, binding to its receptor in the ER or sarcoplasmic reticulum (SR) and triggering intracellular calcium release. DAG activates protein kinase C (PKC), which phosphorylates targets such as TRPV1, a non-selective membrane cation channel, contributing to further intracellular calcium accumulation.

The activation of calcium signaling was supported by the remaining pain cluster hits, albeit acting via different signaling pathways. Three hits in the transcriptome dataset represented venom nerve growth factor 1 (vNGF), which binds a receptor tyrosine kinase (RTK) and directly activates either PLC or PKC (Figure 3). NGFs can activate extracellular signal-regulated kinase (ERK), a transcriptional regulator with key roles in several pain states [31]. Nine additional HCS hits activate protein kinase A (PKA) via GPCRs, and subsequently release calcium from the ER via the ryanodine receptor RyR2 (Figure 3). Intracellular calcium modulation was also supported by HCS hits on voltage-gated calcium channels and TRPA1, a non-selective sodium/calcium channel. However, six HCS hits provided evidence for the depletion of intracellular calcium, although the subsequent calcium-modulated reactions include the induction of metabolic processes, muscle contraction, exocytosis, autophagy, apoptosis, neurogenesis or neuronal transmission [32,33]. EF-hand proteins are involved in several of these processes, and our transcriptome dataset included 11 additional hits for the calmodulin homolog calglandulin.

Our novel network pharmacology approach has a remarkable multiplexing capacity. Depending on the focus, multiple clusters can be built from our integrated shared core functional network, including cardiovascular and hemostasis clusters (Appendix D, Figure A2 and Figure A3). Briefly, the cardiovascular cluster included 14 HCS reference drugs and 17 transcriptome hits against putative cardiotoxins: one endothelin, one cholecystokinin, 15 (acidic/basic) PLA2 homologs, and a natriuretic peptide as another candidate although it was below our expression level cutoff. The highlighted MoA in this cluster indicated the activation of vascular and probably cardiac contraction and relaxation phases via the PKA, PKC, or PKG pathways, which subsequently activate or inhibit myosin heavy and light chains. In the hemostasis cluster (Figure A3), we identified nine hemotoxins with potential fibrin(ogen)olytic activity, including two venom lectins, a serine proteinase inhibitor, a venom plasminogen, a thrombin-like enzyme, and four (acidic/basic) PLA2 homologs.

## 3. Discussion

Animal venoms are a rich source of potent and diverse new drug candidates that could provide large insights into novel MoAs and synergistic activities.

Stingray venom is typically characterized by gel electrophoresis, zymography, in vivo assays or mass spectrometry, usually focusing on freshwater species from the genus *Potamotrygon* [26,34,35]. Transcriptome analysis has also been carried out for three *Potamotrygon* and one *Neotrygon* species [29,30,36]. These studies reported the presence of enzymes, but also other proteins and peptides in the mass range 1–276 kDa. Stingray venom is transferred via a serrated tail spine, causing severe injuries and typical fish envenomation symptoms: intense pain, hemorrhage, edema, erythema, hypotension, secondary necrosis, and infection [34,37]. Our transcriptome and HCS data confirmed the general compositional and functional properties discussed above, including apparent inflammatory, tissue and hemostasis disrupting activity.

The high abundance of transcripts representing the TCTP and hyaluronidase families in our transcriptome datasets agrees with previous findings in *Potamotrygon* species and is commensurate with the symptoms of stingray envenomation [33,34]. TCTP toxins are thought to induce edema [38], whereas hyaluronidases act as venom spreading factors and have been found in two *Potamotrygon* species [26,39]. Interestingly, the putative hyaluronidases we identified were expressed at high levels in both marine and freshwater species, contrasting with the absence of hyaluronidase activity in marine stingray *Dasyatis guttata* [26] and hyaluronidase sequences in the *Neotrygon kuhlii* venom proteome [36].

Stingray venom modulates hemostasis by delaying coagulation via the activity of fibrin(ogen)olytic enzymes [40]. Accordingly, we identified serine proteases and metalloproteinases in all our transcriptome datasets, supporting the bioactivity data and providing the basis for the hemostasis cluster (Figure A2). The abundance of PLA2 transcripts also agrees with the reanalysis (for comparability) of the *Potamotrygon* transcriptomes reported previously (Appendix A) [29,30] and corresponding enzymes have been identified in nearly all animal venoms analyzed thus far. However, only minimal PLA2 activity was found in stingray venom in one previous study [26], and another study claimed that PLA2 has never been detected in fish venom [21]. A possible explanation for this discrepancy is the presence of catalytically inactive K49-PLA2 isoforms as found in snake venoms, which would generate transcriptome hits even in the absence of activity [41]. However, our transcriptome hits solely indicated the presence of active D49-PLA2 (Appendix E, Figure A4) and the role of this enzyme family therefore requires further investigation.

Our transcriptome analysis revealed several intriguing results. First, the augerpeptide was present in all of our transcriptome datasets and was often the most abundant transcript, supporting its detection in the first *P. motoro* transcriptome study [30]. However, the activity of this peptide remains unknown. Second, calglandulin was identified in our transcriptome datasets and in three previously reported transcriptomes representing the genus *Potamotrygon* [29,30], but calglandulin was previously suggested to be a component of the venom secretion process rather than the venom itself [42]. However, the importance of calcium signaling during envenomation and the activation of EF-hand proteins by calcium may indicate that calglandulin has additional roles in the venom that require further investigation. Finally, we identified abundant, high-ranking transcripts representing several neurotoxins of the snake TFT family in our five stingray transcriptomes and in the reanalysis of existing *Potamotrygon* transcriptome data [29,30], supporting the presence of this family even if the expression level is typically low (Appendix A). For further clarification of these hits, we assessed our transcripts for the characteristic TFT patterns (i.e., cysteine scaffold, TOLIP domain, and the absence of certain post-translational modifications) and found that at least 10 transcripts fulfilled all criteria and thus confirm the presence of TFT-like proteins in stingray venom (Appendix E, Figure A5). Despite a number of studies indicating a stronger relationship between the transcriptome and proteome, in terms of presence/absence if not abundance [43], the augerpeptide, calglandulin and TFT hits, as well as the PLA2 transcripts and the marine hyaluronidases, require further confirmation at the protein level.

Our bioactivity assay revealed compartmental signals that provide insight into the cellular responses to stingray venom. Although the precise MoAs remain unknown, the translocation of Nf-κB, mitochondrial impairment, and the disruption of the ER and lysosomes indicate oxidative stress at the cellular level [44]. The mitochondrial signal in particular matches previous finding that venom of the marine stingray *Pastinachus* (*Dasyatis*) *sephen* alters mitochondrial membrane potential by enhancing the production of reactive oxygen species [45]. In the ER and lysosome, swelling, loss of membrane integrity, and even toxin accumulation can contribute to the observed effects [44,46,47,48]. Another scenario for the loss of organelle and plasma membrane integrity is the observed degradation of the cytoskeleton triggered by *P. violacea* venom, as also reported for cnidarian toxins [49]. The loss of plasma membrane integrity may explain the cytotoxicity of marine *P. violacea* crude venom in our bioactivity assay. Contradictorily, it appears that the *Potamotrygon* crude venom did not induce cell death on our bioassay, although necrosis in freshwater envenomations is typically remarkably severe [26,34]. Given the inferred importance of calcium signaling as a target of stingray toxins, the disruption of the ER, mitochondria, and lysosomes may reflect their role as intracellular calcium stores [33].

Each dataset provided only limited insight into the MoAs and synergistic activity of stingray venom components, so to achieve a more holistic interpretation we integrated the gene expression and bioactivity data in a novel network pharmacology approach. Several fields, including natural product research, drug discovery, and pharmacology, typically involve large-scale screens based on target binding or MoAs [50], but more recent trends combine these approaches to provide data on compound-target interaction as well as on-target and off-target MoAs [51,52]. Special emphasis is placed on the identification of key nodes that modulate system phenotypes in the context of multi-target drugs and drug combinations [18]. Our approach adds to the growing number of studies addressing this knowledge gap. The resulting data network revealed that the functional integration of omics data can identify MoAs and potential synergistic interactions for multiple system perturbations visualized as clusters for major stingray envenomation symptoms.

The primary effect induced by stingray venom is pain. The induction or modulation of peripheral pain by animal toxins is often linked to ligand-gated or voltage-gated ion channels, such as TRPV1 and Nav1.7, respectively [53]. However, our pain cluster suggested that the major pain-inducing mechanism of stingray toxins is the activation of calcium signaling via GPCRs. Three major GPCR-dependent pain pathways are known—the excitatory IP3 and PKA pathways, and the inhibitory PKG pathway [54]—and all three were highlighted by our pain cluster. However, the main route synergistically activated by two stingray venom transcripts and supported by 16 HCS hits appears to be the GPCR-IP3 pathway. Several lines of evidence indicate that GPCRs in nociceptors (pain-perception neurons) are targeted by venom proteins, especially those from marine animals [54]. For example, sea anemone gigantoxin I activates the GPCR-IP3 cascade, followed by the phosphorylation of TRPV1 and calcium release [55]. Stonefish verrucotoxin appears to act as a β1-adrenoceptor agonist, increasing L-type calcium currents via the PKA pathway and inducing neurotransmitter release [56]. Furthermore, the synthetic ω-conotoxin ziconotide (an approved drug) and two α-conotoxins from the cone snail block N-type calcium channels in response to inhibitory GPCR signaling [57,58].

The perception of severe pain may involve the combined activation of several pathways [59]. In our pain cluster, we identified multiple targets that activate calcium signaling via different routes (IP3, PKA, and also by directly acting on ion channels). However, we also found evidence for alternative routes, such as NGFs that induce pain via RTK signaling and downstream ERK and IP3 pathway activation [31]. This is often related to inflammation, another well-documented stingray envenomation symptom. The multiple predictions of serotonin and histamine receptor activation in our pain cluster support the intimate relationship between inflammatory mediators and nociceptor hypersensitivity, sometimes leading to hyperalgesia [59]. Nociceptors can be activated directly by venom-related histamine, bradykinin or serotonin, as previously suggested for stingray venoms, or toxins mimicking their activity [60,61]. However, indirect activation is also possible when venom compounds induce the massive release of endogenous inflammatory mediators, for example by mast cell degranulation [60]. The abundant TCTP transcripts in our datasets may encode key mediators of this process by strongly inducing the release of histamine [38], although the precise mechanism is unknown. The synergistic activity of TCTPs and hyaluronidases may further trigger the severe necrosis often associated with stingray envenomation [34]. Necrosis is often more severe following envenomation by freshwater stingrays, and we congruently identified nine HCS hits for HRH1 that were exclusively related to the bioactivity profile of *P. leopoldi*.

We also found strong evidence for the polypharmacology of stingray venom components. The candidate pain-inducing toxins in our pain cluster were bibrotoxin and cholecystoxin, which are associated with neuropathic pain and inflammation but also cardiovascular disorders [62,63]. Accordingly, these candidates were not only identified in the pain cluster, but also in the cardiovascular cluster (Figure A2). Their activity in each cluster may involve different second messengers, but ultimately cause the release of intracellular calcium thus matching our prediction of IP3-calcium pathway activation.

Taken together, our results agree with previous network pharmacology data showing that multi-omics data integration can provide robust predictions of MoAs and synergies in animal venoms that have not been characterized in detail. Our transcriptome and bioactivity data indicate that stingray venom includes enzymes, other proteins and a small number of peptides that can induce pain, disrupt the tissue matrix and hemostasis, and induce pro-inflammatory and cardiotoxic activity. Our integrative approach relies on the quality and abundance of functional data as well as additional manual curation. However, the integrated data network and resulting clusters comprehensively unraveled the mechanisms underlying major stingray envenomation symptoms and their timeline. First, pain is mainly induced via GPCR-IP3 signaling and potentiated by GPCR-PKA, NGF-ERK/IP3, and voltage-gated ion channel activation. Second, inflammation triggers further pain based on the massive release of intracellular histamine, for example by strongly expressed TCTP toxins. Finally, tissue disruption leads to the typical necrotic profile probably reflecting the combination of strongly expressed hyaluronidases and massive inflammation. Our network pharmacology approach was therefore able to identify several routes via which stingray venom synergistically induces system perturbations, but we also found initial evidence for the polypharmacological nature of stingray toxins that trigger multiple systems such as nociception, cardiovascular and immune.

## 4. Materials and Methods

### 4.1. Crude Venom Extracts and cDNA Library Preparation

Venomous tissue (45–212 mg) was scraped from the tail spine of mature live specimens of two freshwater stingrays (*P. leopoldi* and *P. motoro*; Potamotrygonidae) and three marine stingrays (*D. pastinaca*, *H. leoparda* and *P. violacea*; Dasyatidae) following Directive 2010/63/EU on the protection of animals used for scientific purposes. Total RNA was extracted for mRNA isolation, cDNA library construction and sequencing (50 million paired-end reads) carried out by Vertis Biotechnologie (Freising, Germany) and Macrogen (Amsterdam, Netherlands) using the Ilumina HiSeq2000 platform. For high-content bioactivity screening, crude venom was obtained from 30–104 mg of spine tissue from mature live specimens of *P. leopoldi*, *D. pastinaca* and *P. violacea* using a standardized methanol-based bioactive compound extraction protocol (Appendix C, Table A2).

### 4.2. Venomous Tissue Transcriptome Analysis

The transcriptome analysis workflow is shown in Figure A1. Briefly, 182 Gb of sequence data from five new stingray transcriptomes (species listed above, one specimen per transcriptome) and three existing datasets [29,30] were quality checked, trimmed and assembled using a combination of Trinity [64] and rna-SPAdes [65] to reduce algorithm-specific peculiarities (Table A1). Gene expression levels were calculated by mapping raw reads back onto the assembled transcripts using HISAT2 [66] and sequences with low coverage (TPM < 0.5) were discarded. Translated transcripts were annotated by sequence comparison in BLAST [67] and HMMER [68] using UniProt [69] and Pfam [70] as references, followed by venom-related annotation solely against ToxProt [71]. Annotations were filtered by identity (≥40%), coverage (≥40%), and bit score (≥30) and only the highest-scoring hit of the remaining transcripts against a unique UniProt ID was taken into account. Putative toxin family classification was based on ToxProt and was collapsed at the superfamily level where possible. Toxin family expression is presented as relative values (the sum of the expression values of single hits in a single toxin family divided by the number of these hits). Transcripts were further annotated against KEGG [72] and KEGG Medicus [73] to improve cross-linking to the bioactivity data and to identify signaling pathways and target systems. Functional category names from KEGG are herein denoted with capital and italics.

### 4.3. High-Content Bioactivity Screening (HCS)

We compared the effects of stingray venom extracts to the effects induced by 712 reference compounds from the LOPAC 1280 library (Sigma-Aldrich, Steinheim, Germany) in a HeLa cell bioactivity assay. Briefly, serially-diluted venom extracts were tested in four replicates. After 24 h, four fluorescence-based cell-staining protocols (Appendix C) were used to stain nine cellular targets (Table A3). HCS was carried out on the Cellomics ArrayScan VTI platform (Thermo Fisher Scientific, Waltham, MA, USA) equipped with a 10x objective EC Plan Neofluar (Zeiss, Oberkochen, Germany). Images were analyzed using the Compartmental Analysis Bio Application (Cellomics). At least 500 valid objects were analyzed per well. Cell loss and cycle analysis was carried out in parallel using the Cell Cycle Bio Application (Cellomics) and a minimum of 2000 valid objects. Raw data were processed for stepwise data reduction and statistical evaluation until a final set of 31 highly informative parameters remained.

For each reference compound and crude venom sample, the induced cell responses were aggregated into high-resolution cytological profiles, defined as unique fingerprints of cellular perturbations generated by joining distinct features derived from spatially-resolved measures of fluorescence intensities relative to control treatments [74]. These profiles were processed by hierarchical clustering (complete linkage clustering and Spearman rank correlation) in Multi Experiment Viewer v4.9 [75]. Clusters were defined as compounds (venom-associated or references) that fell within a distance threshold of 0.7. We assumed that similarities in phenotypic responses among compounds indicate similar biological targets and MoAs [52,76]. Therefore, as described above for transcriptome analysis, reference compounds clustering with crude venom profiles were manually annotated against KEGG, linking the phenotypic responses to the transcriptome-based activity profile.

### 4.4. Integrative Approach

The functional data inferred independently by transcriptome analysis and bioactivity screening were integrated to create a dataset of shared signaling pathways including functionally annotated hits generated by both approaches. From this integrated dataset, single clusters were identified for each envenomation effect, as herein, but can also be identified for target systems and toxin families/classes. The cluster for the primary stingray envenomation effect (pain) was created by combining all pathways potentially related to this effect and using the hits in these pathways to reveal potential molecular targets and induced cellular responses either by functional annotation in KEGG or additional manual curation. Finally, the venom transcripts, molecular targets and cellular response data were compiled in specific clusters to identify drug candidates in the stingray venom and predict their major MoAs.

## Figures and Tables

**Figure 1 marinedrugs-20-00027-f001:**
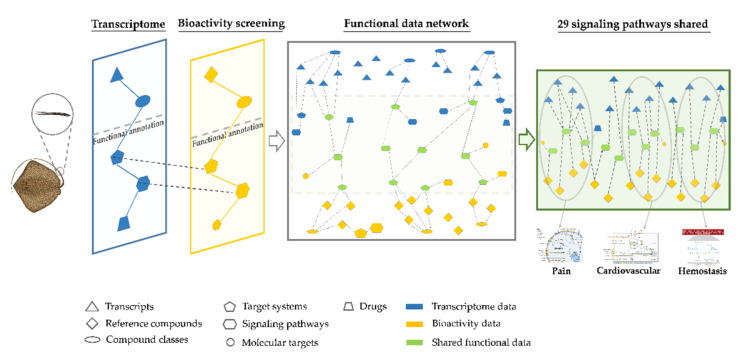
Schematic representation of the integration workflow used to predict the mechanisms of action for drug candidates in crude stingray venom. First, venomous spine tissue from five stingray species was used for transcriptome analysis and high-content bioactivity screening. Second, the functional data inferred from both approaches were integrated, enabling the identification of clusters specific for particular envenomation symptoms that reveal candidate toxins and mechanisms of action. The spine 3D model was reproduced with permission from Dr. Jessica Reichert and the stingray image was based on a photograph taken by Hamid Badar Osmany (www.fishbase.org, accessed on 6 August 2019).

**Figure 2 marinedrugs-20-00027-f002:**
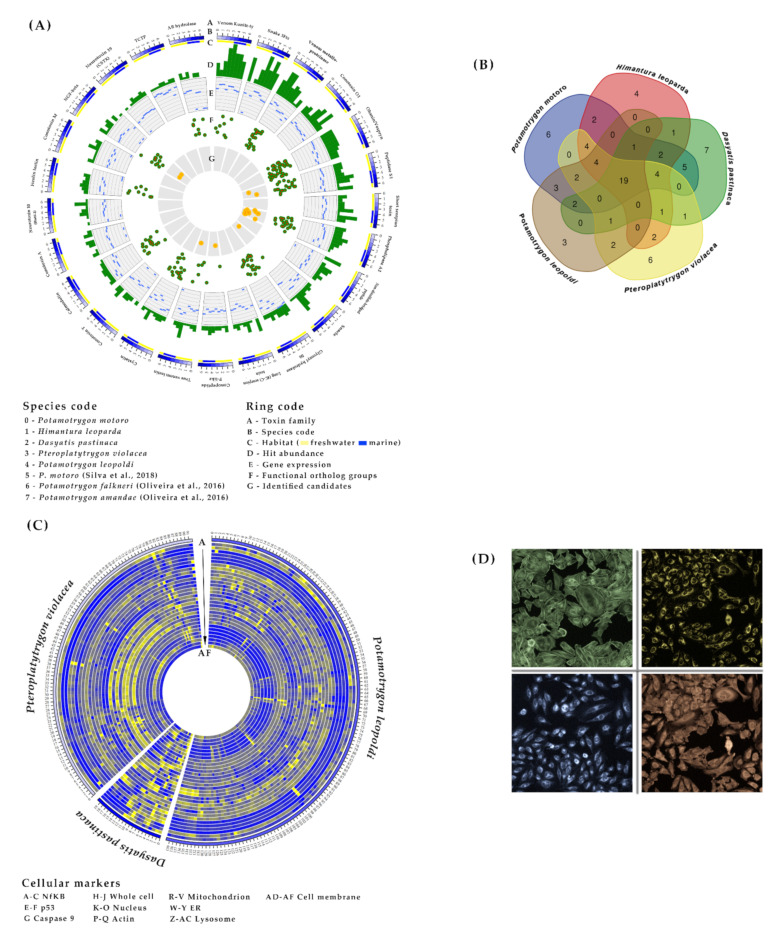
Putative venom composition and venom bioactivity profiles of stingray species. (**A**) The putative venom composition inferred from eight transcriptome datasets (five newly analyzed, and three based on publicly available data denoted in the species code with number 5–7 [29,30]) from seven stingray species, is represented by the top 25 toxin families. (**B**) A Venn diagram based on the toxin family data reveals interspecific similarities. (**C**) The venom bioactivity profile obtained from a high-content screening assay in HeLa cells is represented as a heat map, including 31 highly-informative cytological parameters and 712 reference compounds clustering with stingray venom extracts within a distance threshold of 0.7 (cytological parameters, from outer to inner, are denoted by the letter codes A-AF, which are listed in the Appendix C). (**D**) Images of stained actin (green), ER (blue), Nf-κB (red), and mitochondria (yellow) in HeLa cells.

**Figure 3 marinedrugs-20-00027-f003:**
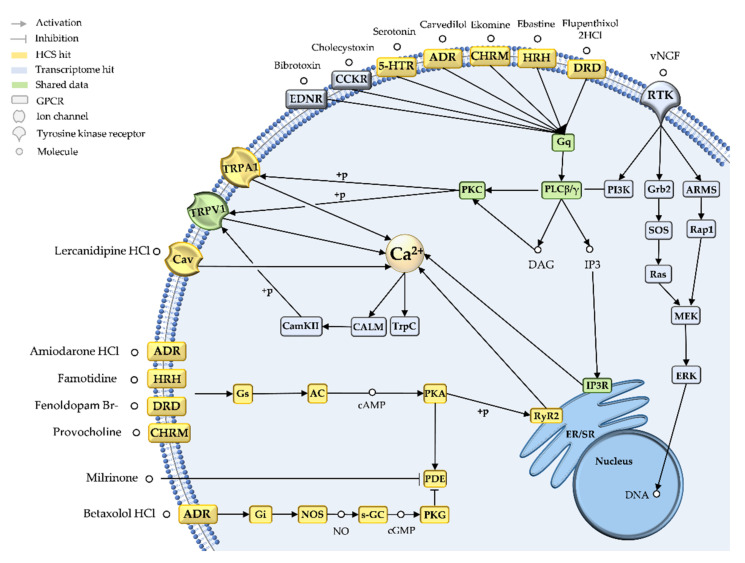
Pain cluster based on integrated transcriptomic and bioactivity data. The integration of transcriptomic (blue icons) and high-content bioactivity screening data (yellow icons) predicts the mechanism of action of pain-related drug candidates in stingray crude venom. Green icons show consistencies in both datasets.

## Data Availability

Transcriptome datasets analyzed in this study are publicly available and can be found here: https://www.ncbi.nlm.nih.gov/sra under accession numbers SRR1104920(4)-8 (BioProject number: PRJNA605588), SRR2039259, and SRR7994305, and https://db.cngb.org under accession number CNP0000235 (the latter three transcriptomes were last accessed on 8 May 2019) The raw high-content bioactivity data presented in this study are available in Appendix A.

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
