# Peer review of "Stingray Venom Proteins: Mechanisms of Action Revealed Using a Novel Network Pharmacology Approach"

_marinedrugs, 2021, doi:10.3390/md20010027_

Round 1

Reviewer 1 Report

The authors are describing an interesting take on assessing the mode of action of stingray venom. Venoms are a valuable source of bioactive molecules often used as pharmacological tools or therapeutics drug leads. However, understanding the mode of action of the venom is still challenging. By combining transcriptomics and high content screening, the authors are carrying out a systematic review of stingray venom and they were able to accurately predict mode of action and synergies in previously undescribed venoms. Transcriptomics is opening the door to study venomous creatures that are hard to keep, and that present challenges when attempting to acquire the venom material. Combining HCS and transcriptomics can provide a full picture of the venom not previously accessible. The paper is well written and would be of interest to readers of Marine Drugs.

Minor comment:

Figures and Tables – the font in almost all the figures is too small and impossible to read. In Fig 1, the words “pain, cardiovascular, and homeostasis” and almost unreadable, and the legend inside the figure is too small as well. In Fig 2, the legends for panel A and C are too small and the text around the wheel in panel A is unreadable, it is so small. In Fig 3, +P in the cell is too small. Tables A2 and A3, the font is too small, especially in Table A3. Fig A2 could do with increasing the font and the font size in Fig A3 should be increased before publication. Fig A3 looks like a screen shot, which I assume it isn’t, but it needs to be fixed. Fig A4 and A5 are useless as they are now. The font is too small to read and all the work that the researcher has done is wasted due to the small size of the figure. Is there a reason for Fi A4 and A5 to be so small?

Discussion – line 252. “Animal venoms are a rich source of potent and diverse drug candidates that reveal novel MoAs and synergistic activities that are rarely assessed” Please re-write the sentence animal venoms can’t reveal new MoAs unless they are assessed.

Appendix B – CO2 should be subscript on the 2 throughout. All measurements are in µL/mL which is not reproducible unless you have the concentration of all the reagents. Please amend this so the experiments are reproducible.

Reviewer 2 Report

Dear Authors,

I carefully reviewed the MS. It is well written and with interest to readers of Marine Drugs, as well as other researchers related to venomics. I have minor comments:
- Figure 1, the resolution of the text inside the figure is difficult to read. Could increase the font size/resolution?
- References are not in agreement with journal requirements. Add only volume, not the issue number.
